# Community-Based Mental Health Challenges and Implications: Examining Factors Influencing Distress and Help-Seeking Behaviors among Korean American Church Leaders and Members in Greater Los Angeles

**DOI:** 10.3390/ijerph21081094

**Published:** 2024-08-19

**Authors:** Kelly Baek, Christi Bell, Susanne B. Montgomery, Larry Ortiz, Akinchita Kumar, Qais Alemi

**Affiliations:** 1898 Business Center Drive, Department of Social Work & Social Ecology, School of Behavioral Health, Loma Linda University, San Bernardino, CA 92408, USAsmontgomery@llu.edu (S.B.M.); larryortiz@llu.edu (L.O.); qalemi@llu.edu (Q.A.)

**Keywords:** anxiety, Christians, churches, church leaders, community-based mental health, depression, Korean American, mental health promotion

## Abstract

There is limited research on the factors that impact mental distress among Korean American (KA) church leaders even though their unique social situation can create many barriers to seeking mental health assistance. This study compared factors impacting mental distress and help-seeking behaviors between KA church leaders (CLs) and church members (CMs) in the greater Los Angeles area. The respondents (*N* = 243) were mostly female, married, educated, first-generation immigrants with a mean age of 47.9 years (*SD* = 19.7). The Hopkins Symptoms Checklist 10 was used to measure anxiety and depression. Hierarchal linear regressions showed that health status exerted the strongest effect on both anxiety and depression among CLs and CMs. Beyond health status, education (only for depression), informal resource use, and resiliency impacted mental distress scores for CLs. Only resiliency and religious coping predicted depression scores among CMs. To effectively reach this population, community-based organizations and behavioral health specialists should consider collaborating with churches to promote and provide essential mental health support. Our findings also highlight that the needs of church leaders (CLs) and church members (CMs) differ, which should guide the development of culturally tailored interventions that build on the resilience of both groups.

## 1. Introduction

Korean Americans (KAs) are twice as likely to report depressive symptoms than the general US population [1] but are known to underutilize mental health services [2], often using these services as a last resort [3]. Kim [4,5] (2012, 2015) found that approximately 30–49% of first-generation KAs report suffering from depression [4,5], while only 17% utilize mental health services [5,6]. Other studies have found prevalence rates for depression as high as 38% among older KAs [7].

Although it is true that many immigrant populations struggle with mental health concerns, many studies look at mental health outcomes for Asian Americans as a homogenous group when, in fact, Asian Americans comprise approximately 43 subgroups [8]. Each of those subgroups differ in language, cultural norms, socioeconomic status, and experiences of immigration, all of which have been found to contribute to mental health needs, service use, and outcomes [7,9], emphasizing the importance of researching specific Asian American subgroups to understand each groups’ needs, service usage, and outcomes.

Generalized mental health interventions often overlook the unique needs of Asian Americans, including KAs [10,11], or they do not include KAs in standard community-based mental health promotions [10,11,12]. This issue is further exasperated by the community’s insularity and reluctance to engage with organizations or groups outside of the KA community [13,14,15,16,17]. To meet the needs of the community, church leaders often act as proxy, but their level of preparedness is not as high as it should be [13,16,18,19]. In addition, the difference in needs between church members and church leaders has not been explored in prior studies. Therefore, we need to understand the factors that impact between both groups (CLs and CMs) to identify what would best work to meet the unique needs of each group and effectively engage in community-based mental health promotion and practices.

## 2. Factors Impacting Mental Distress and Help-Seeking Behavior

Due to acculturation and immigration, KA studies show an increase in psychological distress in subgroup Asian populations. Previous studies have found that first-generation KAs tend to express a more positive effect as they become more acculturated to American culture [2,20]; however, level of acculturation did not appear to have a significant impact on the level of knowledge about depression or attitudes towards more Westernized treatment options [5]. The processes of immigration and acculturation for many immigrants are stressful processes that can lead to acculturative stress and poor mental health outcomes [21]. Acculturation occurs when a group of people with an established culture are introduced to a new culture that is different to their own and must learn the new culture to adapt to it [22]. This process can be fraught with misunderstanding and conflict, leading to a sense of estrangement between immigrants and the host culture [1]. Barnes and Bennett [23] identified that KA immigrant groups are mostly comprised of people whose parents were born in Korea, speak only Korean, and who strongly identify with Korean culture. The challenges of learning a new language, new behaviors, and separation from social supports can create a greater sense of being socially isolated and increase the experience of loneliness [24], leading to an increase in depressive symptoms in KAs [25].

However, many different types of protective factors have been found to be effective in reducing or safeguarding against an increase in negative mental health symptoms. Protective factors of interest for this study include religious coping, resilience, and social supports. Religious/spiritual coping have been found to be effective protective factors in buffering individuals from the most severe mental health symptoms across cultural and age groups [26]. Religious/spiritual coping can include broad supports such as meaning-making, access to meaningful resources, a greater sense of connection with community, and developing a relationship with a divine entity. In daily practice, these can be broken down to include daily prayer, attending church services and activities, reading scripture, or spiritual guidance [26,27]. According to Lee et al. [6], stressful events experienced by older KAs can lead to an increase in religious activities. This would make sense, as so many KAs are already associated with participation in church and religious activities.

Resilience, often described as a person’s or community’s ability to recover from adversity [28,29], is an effective protective factor against psychological distress [30,31]. According to Babic et al. [32], the qualities that capture resilience include people who tend to look at the positive aspects of life, see things that happen to them as experiences of value, appreciate input from others, have strong support systems, and have insight into their feelings. Resilience has been found to include personal qualities and ecological constructs, including relationships to family, friends, communities, and culture [33]. Lastly, the literature has shown that resilience is something that people can learn to nurture and develop, which leads to improved mental health [32,33].

Social support, defined as being connected to people around them who have helped them when they need help or emotional support [34], has also been found to be a protective factor that can serve as a buffering factor in regard to a person’s mental health concerns [35]. Cohen and Wills [36] conducted a study to answer the question of whether social support could serve as a mental health buffer. These researchers identified that the buffering effect occurs when people feel like they have social support in their lives. Since that time, multiple studies have concluded that social support serves an important role in an individual’s mental health status and plays a particular role for Chinese and Korean Americans [34,35,37,38].

## 3. Korean American Churches and Community-Based Mental Health

KA immigrants face significant mental health challenges [1,4,5,7] and, culturally, have a great reluctance to engage in mental health services [2,3,5,6]. In addition, there is a limited number of providers serving the Asian American community [39,40], further limiting engagement with mental health services. These factors, coupled with chronically high rates of distress, call for a reframing of how mental health is conceptualized in this community. Lee et al. [41] and Shin [17] recommended increased education about the biological basis of mental health conditions and various treatment options, such as medication and/or therapy. However, this continues to put the onus on mental health professionals. Yuri et al. [42] suggested a more ecological approach that emphasizes the use of community-based support systems. Since churches remain central in the lives of KAs, churches could serve as a critical part of that support system for the KA community given that approximately 59% of KAs identify as Christians [43]. Korean churches also often act as pillars of support for the KA community and as access points for spiritual, physical, and mental health needs [44], providing a place and opportunity for people to meet and connect [45]. Increased collaboration between church leaders and leaders within community-based mental health organizations can help provide greater access to the KA community using a holistic framework that can inform more culturally grounded services to be utilized as a gateway to open dialogues about mental health, increase awareness, and ease the stigma against mental health services [41,44,45].

Several studies have also proposed having KA religious leaders serve in a more prominent role in community-based mental health due to their unique position in the community and their impact on mental health literacy [41,42]. Church leaders, particularly pastors, are often perceived as the first line of support when community members seek help for social, financial, and mental health challenges [46]. This can present multiple challenges, as clergy can struggle to identify which role is most appropriate to help their parishioners—the clerical or the counseling role. Moreover, even though clergy are provided some training in seminary, clergy are not automatically provided with specialized training as clinicians unless they seek it out [47]. Moran et al. [47] explained that, while clergy felt a great ability to help with more traditional issues, including grief, loss, and marital issues, they felt less capable to deal with the more complex mental health issues that their congregants experienced.

While pastors are called on to assist their congregants with mental health issues, the aspects of distress among church leaders, factors influencing mental distress and help-seeking behaviors, and whether these might differ from those of church members have not been examined. Although the literature is sparse regarding the relationship between religiosity and mental health-seeking behaviors, there are studies that have shown that individuals with high levels of religiosity were more likely to delay or avoid seeking mental health services [48,49]. In reviewing the literature, church leaders’ beliefs about seeking their own mental health care were not found. However, the literature about KAs’ perceptions of stigma related to mental health services [50,51,52] provides a logical path to understanding that KA church leaders might be reluctant to seek mental health services because openness about their mental health struggles may harm their trustworthiness in the community, with parishioners perceiving them as incompetent. This limits church leaders’ willingness to access professional mental health services and also limits the social support (e.g., from friends and family) they can depend on when dealing with mental distress [53]. A lack of culturally sensitive resources and the stigma surrounding mental illness [1] has been found to increase the risk of burnout. Church leaders also fear jeopardizing their reputations and standing in the community, thus creating a significant barrier to seeking help [54]. This may lead to some church leaders utilizing fewer public means of alleviating distress, such as praying [55,56] or exercise [57]. In addition, factors such as resilience [28,29] and religiosity [57] can act as protective buffers against mental distress.

When analyzing the factors that influence mental well-being, attention must be paid to intersections and inequalities related to the different social roles of the affected individual at the individual and community level. Intersectionality points to how fundamental factors of inequality (i.e., race, gender, class, religious beliefs, and sexuality) mutually define and reinforce one another [58]. Examining the intersection of these roles at the individual and community level can provide insights into how to develop effective engagement strategies, resources, and interventions that meet the unique needs of church leaders and church members and promote community-based mental health.

Therefore, the purpose of this study was to (1) explore and compare the scope of mental distress (anxiety and depression) and factors that impact mental distress and help-seeking behavior (health status, acculturation, informal resource utilization, religious coping, resiliency, and social support) among KA church leaders and church members, as well as to (2) identify factors that could promote and inform the development of community-based culturally grounded services for service providers and community organizations.

## 4. Methodology

### 4.1. Participants and Procedures

To assess levels of mental distress and factors impacting mental distress and help-seeking behavior among KA church leaders and church members, a cross-sectional quantitative study was conducted in KA churches in the greater Los Angeles area, which has the largest Korean population in the Unites States [59].

The survey included questions about socio-demographic variables, perceived health status, self-reported mental distress, levels of acculturation, informal resource utilization, religious coping, resilience, and social support.

The PI contacted church leaders (i.e., pastors and elders) at Korean churches in the greater LA area. If the church was interested, the PI would visit the church to make a recruitment announcement during the church announcements and interested participants were either meet with the PI after the service or would leave contact information. Adult participants aged 18 and over who identified as having Korean heritage were recruited at seven Korean churches and given the option of completing an English or Korean language hardcopy or online Qualtrics survey. For each survey submitted, a USD 5 donation was made to a church of the participant’s choosing.

Convenience and snowball sampling techniques were utilized to recruit 243 participants of both genders, varying ages, educational levels, and generational status. Of the 243 participants, 110 self-identified as being a formal or elected church leader (i.e., pastors, elders, deacons/deaconesses) and 133 identified as church members. Data were collected from November 2017 to February 2018 from seven churches of various denominations (Seventh Day Adventist, Presbyterian, Non-denominational, Methodist, Catholic, and Baptist) and sizes (150 members to 1000+ members).

The study was approved by the Loma Linda University IRB on 14 November 2017 (protocol code 5170318). Written informed consent was obtained from all individual participants included in the study, and they were informed that their participation was voluntary and that they could withdraw at any time. Participants were also informed in writing that their de-identified data could be used for research and that the data were stored on secure equipment (e.g., password protected computers) and facilities (e.g., locked file in a locked room in a secure building.

### 4.2. Measures

For the purpose of this study, mental distress was defined as depression and anxiety. In addition, the survey clearly defined what type of distress the questions pertained (e.g., it was explained that the Hopkins Symptoms Checklist was asking about frequency of depression and anxiety symptoms within the last week).

#### 4.2.1. Dependent Variables

Anxiety and Depression. The Hopkins Symptoms Checklist (HSCL) 10 is a 10-item self-reporting symptom inventory. The first four questions measure symptoms of anxiety and the final six questions measure depression [60]. Responses to each question are based on a 4-point Likert Scale (1: not at all to 4: extremely), with higher scores (ranging from 4–16 for anxiety and 6–24 for depression) indicating greater emotional distress. The mean scores were utilized to assess the level of distress for anxiety and depression. Internal reliability for anxiety (Cronbach’s *α* = 0.81) and depression (Cronbach’s *α* = 0.75) were within the acceptable range. This scale has previously been validated among KAs [3].

#### 4.2.2. Factors Impacting Mental Distress

Acculturation. To measure acculturation for this study, the Short Acculturation Scale for Koreans (SAS-K) was utilized. This 12-item scale consists of measures a person’s acculturation level through three subscales, (a) language (5 items), (b) media (3 items), and (c) ethnic–social relations (4 items), and asked question such as, “In general, what language do you speak” and “Your close friends are”. The responses, based on a 5-point Likert-type scale, ranging from 0 points (only/all Korean) to 4 points (only English or all non-Korean), were averaged across items (range of scores is 0 through 4). Scores ranged from 0–48, with higher scores representing higher levels of acculturation [61]. This scale was validated among Korean adults (2011) and demonstrated excellent internal reliability for this sample (Cronbach’s alpha = 0.94).

Informal Resources. Participants were asked to indicate, on a checklist of coping strategies, how they coped with mental distress in the previous 12 months. This checklist included the following: dealt with it myself, prayed, ignored it, exercised, confided in my pastor/spiritual leader, and/or confided in my family/friends. Each coping method was dichotomized into 0 (No) and 1 (Yes).

Religious Coping. The Brief Religious Coping scale [62] is a seven-item measure of the use of religion to cope with major life stressors. It is divided into three scales. The first scale, composed of three items, measures positive attitudes toward God; the second scale, composed of three items, measures negative views of God; and the third scale measures the extent to which religion plays a part in understanding or dealing with stressful situations, with answers ranging from 0 (very involved) to 3 (not involved at all). While the scale is typically scored according to the aforementioned subgroups, for this analysis all seven items were collapsed (with all items in the second scale reverse coded) to create the religious coping scale, with higher scores indicating a more positive outlook on God and the greater involvement of religion in decision making. Scores ranged from 0 to 19. This scale has previously been used and deemed a good fit for KAs [63]. The scale was translated into Korean by a bilingual native Korean speaker and then back-translated into English by a bilingual native English speaker. The internal reliability for the scale was within the acceptable range (Cronbach’s *α* = 0.72).

Resilience. The Conner–Davidson Resilience Scale (CD-RISC) operationalizes resilience as “the ability to thrive in the face of adversity”. It assesses personal competence, high standards, tenacity, trust in one’s instincts, tolerance of negative effects, strengthening effects of stress, positive acceptance for change and secure relationships, and control. This scale consists of 10 items, each of which is rated by respondents on a 5-point scale (0 = not true at all to 4 = true nearly all of the time) according to the extent to which respondents agree that an item has applied to them over the previous month. Items include statements such as “I believe that I can achieve my goals, even if there are obstacles” and “I am able to adapt to change”. All responses were summed to achieve a total score for each respondent, with higher scores reflecting greater resilience in a range from 0 to 40 [64]. Both the English and Korean versions have been shown to be reliable and valid among Korean adults [65] and demonstrated excellent internal reliability for this sample (Cronbach’s *α* = 0.94).

Social Support. The Lubben Social Network Scale (LSNS) 6 [66] is a six-item scale used to measure perceived social support received from family and friends. The total score ranges from 0 to 30, with higher scores indicating more social engagement with family and friends. The Korean revised LSNS 6 (K-LSNS-6) has been shown to have good internal reliability among KAs [67]. The internal reliability for this scale was within the acceptable range (Cronbach’s *α* = 0.85).

### 4.3. Control Variables

Demographic Variables. We assessed respondents’ gender (female/male), age, marital status (not married/married), educational attainment (high school or less, some college/associate/technical training or more), employment status (yes [including self-employment]/no), and if respondents had health insurance (yes/no).

Income Status. To measure income, a proxy measure informed by Song et al. [68] was utilized. Participants were asked if they could comfortably pay their monthly bills, responding with either yes (1) or no (0).

Generational Status. Participants were asked in which country they were born and, if born in another country, what year they moved to the U.S. Participants who moved to the U.S. before the age of 12 years or who were born in the U.S. were categorized as 1.5 or 2nd generation, respectively (0), and those who were born in another country and moved to the U.S. after the age of 12 years were categorized as 1st generation (1).

Physical Health. A single item from the Short Form Health Survey 12 (SF 12) was used to measure perceived physical health. Respondents were asked, “How would you rate your health?” with a possible response ranging from poor (0) to excellent (5) (reverse-coded), providing a general perception of overall health and values for health-related quality of life [69].

### 4.4. Data Analysis

SPSS, version 29.0 (Manufacturer: IBM Technology Company; released September 2022; Armonk, NY, USA), was used for all data analyses. Prior to conducting any analyses, data were tested for missing data, outliers, and violations of assumptions. A total of 27 surveys with more than 10 percent missing data overall were excluded from the study. The data were also assessed via scatterplots to determine what type of data were missing. Missing data were found to have 5 percent or less missingness for each participant and for each scale. In addition, the pattern of missingness was random. Therefore, mean imputation was used for continuous variables to optimize data for the analyses [70].

Frequency and descriptive analyses were run for all socio-demographic variables as well as protective factors for each distress scale. Descriptive statistics were generated for all socio-demographic variables as well as for all the scales. Bivariate analyses (independent samples *t*-tests and Pearson’s correlation) were conducted to (1) examine the existence of differences in distress scores between church leaders and church members and (2) examine the existence of differences in relationships between distress scores and other factors between church leaders and church members. Hierarchal linear regression was used to build four models pertaining to factors that predicted anxiety and depression scores for each group.

## 5. Results

### 5.1. Participant Characteristics

The sample population (*N* = 243) was mostly female, married, had attained some college education or more, and was first generation. Respondents ranged in age from 19 to 90 years, with a mean age of 47.9 years (*SD* = 19.7). Most participants were employed, able to comfortably pay their monthly bills, and had health insurance. Church leaders (*n* = 110) were predominantly female (59.1%) and more likely to be older than the average overall population, with ages ranging from 24 to 84 years and a mean age of 55.9 years (*SD* = 16.18). They were also more likely to be first generation (78.2%), married (81.8%), have attained some college education or more (89.1%), not employed (51.8%), be able to comfortably pay their monthly bills (86.4%), and have health insurance (86.4%). Most church members (*n* = 133) reported being female (61.7%), 1.5/2nd generation (57.1%), being not married (60.2%), having attained some college education or more (91.0%), not employed (51.1%), being able to comfortably pay their monthly bills (75.9%), and having health insurance (89.4%). The average mean age of church members was 41.22 (*SD* = 19.96), ranging from 19 to 90. See Table 1 for socio-demographic information for each group.

### 5.2. Bivariate Results

Independent samples *t*-tests were run to compare the level of distress symptoms between the groups. The mean scores for anxiety were similar between church leaders (*m* = 1.55, *SD* = 0.55) and church members (*m* = 1.53, *SD* = 0.50), *t*(226) = 1.26 *p* = 0.21. However, depression scores for church members (*m* = 1.75, *SD* = 0.47) were significantly higher than that of church leaders (*m* = 1.59, *SD* = 0.50), *t*(241) = −2.50, *p* = 0.01. It should be noted that only health status and resiliency were significantly associated with both distress variables for both groups. A key difference was that use of several of the informal resource utilization categories and social support were significantly different for church leaders yet were all non-significant for church members. In addition, more socio-demographic variables, such as age, generational status, and marital status, were significantly different for church members but not church leaders. See Table 2 for full bivariate results.

#### 5.2.1. Church Leaders

The only socio-demographic variable that was significantly related to mental distress was educational status, with those with higher educational levels (*m* = 1.63, *SD* = 0.49) reporting greater frequency of depressive symptoms, *t*(108) = −2.24, *p* < 0.05. Poorer perceived physical health was also correlated with higher anxiety (*r*(108) = −0.39, *p* < 0.001) and depression (*r*(108) = −0.32., *p* < 0.001) scores. Regarding coping strategies, those that reported dealing with the distress by themselves had significantly higher scores for both anxiety (*m* = 1.77, *SD* = 0.62), *t*(108) = −2.52, *p* < 0.01 and depression (*m* = 1.78, *SD* = 0.52), *t*(108) = −2.61, *p* < 0.01. Individuals who also reported that they ignored their mental distress were also more likely to report higher depression scores (*m* = 1.97, *SD* = 0.44), *t*(108) = −2.01, *p* < 0.05. In contrast, church leaders that reported that they prayed, (*m* = 1.49, *SD* = 0.42), t(108) = 3.38, *p* < 0.001, and confided in a pastor or a spiritual leader, (*m* = 1.33, *SD* = 0.31), *t*(14) = 2.59, *p* < 0.05, had significantly lower depression scores. Greater levels of resiliency were also correlated with lower anxiety (*r*(108) = −0.36, *p* < 0.001) and depression (*r*(108) = −0.41, *p* < 0.001) scores, while more positive religious coping (*r*(108) = −0.27, *p* < 0.001) and higher levels of social support (*r*(108) = −0.19, *p* < 0.05) were associated with lower depression scores. Age, gender, generational status, marital status, employment status, income level, if they had health insurance or not, exercising, confiding in family and/or friends, and acculturation were not significantly associated with any of the distress variables.

#### 5.2.2. Church Members

Church members who were older, (*r*(131) = 0.25, *p* < 0.001), first generation, (*m* = 1.65, *SD* = 0.55), *t*(131) = −2.23, *p* < 0.05, married (*m* = 1.65, *SD* = 0.55), *t*(131) = −2.12, *p* < 0.05, and had lower levels of education, (*m* = 1.81, *SD* = 0.49), *t*(131) = 1.99, *p* < 0.05, reported higher anxiety scores. Higher depression scores were also reported for individuals who were not able to comfortably pay their bills every month, (*m* = 1.91, *SD* = 0.59), *t*(131) = 2.31, *p* < 0.05. Poorer health status was also significantly correlated with higher anxiety, (*r*(131) = −0.45, *p* < 0.001), and depression, (*r*(131) = −0.31, *p* < 0.001) scores. In contrast, higher resiliency scores were also associated with lower levels of anxiety (*r*(131) = −0.30, *p* < 0.001) and depression (*r*(131) = −0.32, *p* < 0.001), and church members with more positive religious coping also reported lower depression scores (*r*(131) = −0.25, *p* < 0.01). There were no significant relationships between gender, employment status, health insurance, informal resource utilization, acculturation, social support, and the mental distress variables for this group.

### 5.3. Multivariate Analyses

Hierarchal linear regressions were run for each distress variable for each group, controlling for significant socio-demographic variables and health status in the first step. In the second step, the other variables that were significant at the bivariate level were included (significant informal resource use variables, religious coping, resiliency, and social support).

#### 5.3.1. Church Leaders

The regression analyses showed that the variables entered in Step 1 significantly predicted both anxiety (*R*^2^ = 0.21, *R*^2^*_adjusted_* = 0.17, *F*(6, 103) = 4.64, *p* < 0.001) and depression scores (*R*^2^ = 0.21, *R*^2^*_adjusted_* = 0.16, *F*(6, 103) = 4.50, *p* < 0.001), with poorer health significantly predicting higher anxiety (*β* = −0.42, *p* < 0.001) and depression (*β* = −0.37, *p* < 0.001) scores. Step 2 also significantly predicted anxiety (*R*^2^ = 0.34, *R*^2^*_adjusted_* = 0.25, *F*(13, 96) = 3.78, *p* < 0.001) and depression (*R*^2^ = 0.41, *R*^2^*_adjusted_* = 0.33, *F*(13, 96) = 5.16, *p* < 0.001), with poorer levels of health continuing to significantly contribute to the model for anxiety (*β* = −0.34, *p* < 0.001) and depression (*β* = −0.24, *p* < 0.01) as well as higher levels of education predicting higher depression scores (*β* = 0.26, *p* < 0.01). Church leaders who dealt with mental distress by themselves were also more likely to report higher anxiety (*β* = 0.25, *p* < 0.01) and depression scores (*β* = 0.21, *p* < 0.01). Higher levels of resiliency also significantly predicted lower anxiety (*β* = −0.22, *p* < 0.05) and depression scores (*β* = −0.22, *p* < 0.05). Age, generational status, marital status, income, using prayer, ignoring their mental distress, religious coping, and social support did not significantly contribute to any of the models. See Table 3 for full multivariate results for church leaders.

#### 5.3.2. Church Members

For church members, while Step 1 significantly predicted both anxiety (*R*^2^ = 0.24, *R*^2^*_adjusted_* = 0.20, *F*(6, 126) = 6.59, *p* < 0.001) and depression (*R*^2^ = 0.12, *R*^2^*_adjusted_* = 0.08, *F*(6, 126) = 2.83, *p* < 0.05) scores, the only variable that exerted an effect was health status (anxiety [*β* = −0.42, *p* < 0.001]; depression [*β* = −0.30, *p* < 0.01]). Step 2 also significantly predicted anxiety (*R*^2^ = 0.20, *R*^2^*_adjusted_* = 0.20, *F*(6, 119) = 3.60, *p* < 0.001) and depression (*R*^2^ = 0.23, *R*^2^*_adjusted_* = 0.15, *F*(6, 119) = 2.73, *p* < 0.01) scores, with health exerting the greatest effect with anxiety (*β* = −0.36, *p* < 0.001). However, there appeared to be a more equal effect between health (*β* = −0.20, *p* < 0.05), religious coping (*β* = −0.23, *p* < 0.05), and resiliency (*β* = −0.21, *p* < 0.05) for the depression model. Age, generational status, marital status, educational status, income, any of the informal resource uses, and social support did not significantly contribute to any of the models. See Table 4 for full multivariate results for church members.

## 6. Discussion

The purpose of this study was to explore and compare the scope of anxiety and depression among KA church leaders and church members by assessing differences in levels of distress and the factors that influence distress and help-seeking behavior between these two groups in addition to identifying factors that can effectively promote community-based mental health awareness and inform culturally grounded interventions. While it was expected for church leaders to report higher levels of distress, the results showed that there were no significant differences for anxiety scores. In addition, church members reported significantly higher depression scores. This could be explained by church leaders using more informal resources to deal with distress than church members. As a result of the work that church leaders do, they are more likely to engage in more religious coping strategies than the average lay person. It is also noteworthy that we did not find differences between socio-demographic variables, with the exception of educational status, at the multivariate level for either group. This lack of difference may be due to similar levels of anxiety and depression across the majority of socio-demographic characteristics, and it suggests that, while factors influencing mental distress may vary depending on the intersections of roles, there are overall high levels of distress across all demographic subgroups. Of the factors that impacted distress, most variables that predicted distress were unique to each group at both the bivariate and multivariate level.

### 6.1. Socio-Demographic and Health Variables

The only socio-demographic variable significantly associated with depression among church leaders was the level of educational attainment, with higher levels of education predicting higher depression scores. While higher levels of education are usually associated with lower levels of depression, this relationship may be impacted by the intersection of the church leaders’ roles, as the average age was approximately 15 years greater than of church members, with 78% reporting to be first generation, in contrast to 43% of church members.

For church members, while none of the socio-demographic variables significantly predicted any of the distress scores at the multivariate level, age, generational status, marital status, and educational level were all significantly associated with anxiety at the bivariate level, suggesting that social identity may have a greater influence on distress for church members than church leaders.

Physical health was also found to be a significant predictor of anxiety and depression for both groups at the multivariate level. Physical health is broadly known to affect anxiety and depression [71], and the results support that health status could serve as a protective buffer for both types of distress.

### 6.2. Acculturation

In contrast to the literature [2,20,21], acculturation was not significant at either the bivariate or multivariate level for any of the distress types. This may be due to the overall similar levels of anxiety across both groups. For church leaders, this could be due to having strong social support or regularly utilizing the social support they have [34]. While depression scores were higher for church members, it appears that this was driven more by differences in the use of coping mechanisms than acculturation levels. Another explanation is that acculturative stress, as opposed to acculturation, has been found to be a greater concern for mental health concerns in KAs [72].

### 6.3. Informal Resource Utilization

Several types of informal resource utilization were significantly associated with distress at the bivariate level for church leaders, while none were associated with either type of distress for church members on any level. While only “dealing with it myself” remained significant at the multivariate level for anxiety and depression, it should be noted that church leaders that used prayer and confided in pastors or a spiritual leader reported significantly lower depression scores at the bivariate level. In addition, those that reported that they ignored their distress also had higher depression scores. This aligns with the literature, which found that praying [6,26] and clergy use of peer support groups [73] served as protective factors.

### 6.4. Religious Coping

It is of interest that, in this sample of church leaders, religious coping was not associated with distress at the multivariate level, in contrast to church members, for whom religious coping was significantly associated with depression at both the bivariate and multivariate levels. In addition, while prayer was associated with depression in the bi-variable analysis for church leaders, the lack of a role for religious coping-related variables may reflect the fact that the sample was recruited from churches and that the respondents who presented with similarly high levels on these constructs as church leaders were more likely to engage in religious coping than the average lay person due to the nature of their work.

### 6.5. Resilience

Resilience was found to be a significant predictor of anxiety and depression for church leaders and depression for church members. This is of interest, as resilience may be a promising factor that can be incorporated as a self-care mechanism. Resilience has more recently emerged as a protective variable for mental distress [74,75].

### 6.6. Social Support

Social support was significant for depression only at the bivariate level for church leaders, with physical health status, resilience, and informal resource use (i.e., dealing with it myself) exerting a greater effect at the multivariate level. However, it should be noted that church leaders who dealt with the distress on their own were at greater risk of depression, suggesting that being able to depend on a network of friends and family in times of need is especially critical for those at high risk of depression who may have limited access to formal support services. In addition, church leaders who confided in a pastor or spiritual leader reported significantly lower depression scores. These results reflect the findings from the Miles and Proeschold-Bell [73] study that identified peer support groups as effective strategies for dealing with mental distress amongst clergy. However, it is recommended that, when utilizing social support systems in interventions, it should be in conjunction with other foci, such as improving physical health and resiliency.

The marked differences in factors that influence distress among these two groups suggest that subgroup differences should be considered when developing culturally appropriate resources and interventions. The factors predicting anxiety and depression among religious leaders versus congregant members highlight that different interventions are warranted to improve mental health. Understanding the important role of social support, religious coping strategies, and increased connection to church members who share age, generational status, educational, and marital status can be helpful in developing interventions for church members, while focusing on physical health and educational enhancement could prove helpful for church leaders as a means of improving and maintaining overall mental health.

### 6.7. Informing Community-Based Interventions and Mental Health Promotion

As found in this study, and noted in the studies by Leong and Lau [9] and Cheung, Leung, and Cheung [3], the interconnectedness of physical health, resilience, religious beliefs, and mental health underscores a holistic perspective in which the physical, spiritual, and mental realms are intricately linked. This highlights the potential of health, resilience, and religiosity to serve as protective factors against distress in the KA community, suggesting that these three elements should be integrated into culturally tailored resources and interventions. A resiliency-based model that can be adapted or modified to the unique needs of each group, such as the community resiliency model (CRM) [30], could also be helpful in decreasing mental distress. The CRM is a somatically based, non-stigmatizing, trauma informed, and resiliency focused community-based program that teaches people how to re-set the natural balance of the nervous system through a set of six wellness skills [30]. This model has been adapted to align with diverse cultures and communities globally. Also, given its biological approach and stress-reduction orientation, it is non-stigmatizing and has been proven effective in significantly decreasing distress and increasing well-being [76,77,78,79,80,81,82], even in isolated communities [76,78,81,82]. The program’s biological focus is particularly appropriate given that Korean Americans tend to express mental distress through physical symptoms [83]. Capacity can also be built up in the community through this approach by training non-mental health professionals, such as church leaders, who, in turn, can train others to utilize these skills. Offering such psychoeducation through churches can not only promote mental health awareness, reduce stigma, and increase mental health literacy but can also provide spaces for support groups tailored to the specific needs of different groups.

In addition, given the influence of churches and their leaders, it is recommended that mental health organizations collaborate more closely with Korean churches because they can provide strong protective buffers (e.g., provide social support, promote resiliency). Studies support the idea that such partnerships could increase awareness, disseminate information, and help reduce the stigma and distrust surrounding mental health services [19,84]. Establishing community centers at or near churches and offering culturally grounded physical, mental, and spiritual services could further ease the stigma of mental illness and encourage open dialogue about mental health [19]. Additionally, providing mental health training for pastors could empower them to guide their congregants toward seeking help when facing challenges [84], especially those who feel overwhelmed and have exhausted other resources.

### 6.8. Limitations

Since participants were recruited solely from churches in the greater Los Angeles area, this research cannot be generalized to other settings. Further, due to the privacy concerns in the KA community in the areas of mental health research, because of stigma associated with mental illness [85], convenience and snowball sampling was utilized. This type of sampling method also allowed the researchers to tailor recruitment (e.g., engage with trusted community figures to help spread the word about the project, recruit participants from settings where KAs naturally gather, such as churches, providing materials in both English and Korean) and leverage community networks (e.g., build on existing networks to get additional referrals) to help recruit in a timely and cost-effective manner. However, using these sampling methods further limited generalizability.

Another limitation was the diverse and small sample size. The recruited sample represented participants with varying educational and employment backgrounds, religious affiliations, generational statuses, ages and genders. Due to this, there was an increased risk of sampling error (e.g., bias in representation and random variability) and difficulty in analyzing subgroups. It should be noted that the data were pulled from a study that assessed the scope and factors that impact mental distress in the general KA population and was not originally intended to be split into subgroups. While significance was still found at the multivariate level, having a more robust sample and greater numbers in the various categories (e.g., religious affiliation and roles) would have been ideal.

It is acknowledged that the research relied on cross-sectional data; therefore, a true cause–effect relationship cannot be established. However, due the limited literature on the health disparities between clergy and community members, particularly in the KA community, the authors believed that it was justified to obtain a baseline of mental distress in this unique population. It is recommended that, once clear links between exposures and outcomes have been established for this population, future work could consider the element of temporal precedence in more rigorous observational or experimental studies by implementing a longitudinal study with a larger and more rigorous sampling method.

## 7. Conclusions

Overall, the results of this study suggest that distress levels among KA church leaders and members are high. However, there are considerable differences in the factors that impact distress among church leaders and church members, highlighting that church roles should be taken into consideration when developing or identifying much needed culturally aligned interventions that will be optimally effective based on social location.

While this study only examined the impact of informal resource utilization and other potential protective factors, it does not identify or address the barriers for church leaders and formal resource utilization. It is suggested that future research should include determining if there is a difference between formal supportive services and informal supportive services for members of the KA Christian community as well as other faith-based communities. Furthermore, an examination of the cultural considerations needed to design culturally aligned interventions for members of the KA community, as well as other immigrant communities, is needed.

It is hoped that the findings of this study will encourage other church leaders and members to consider whether there are barriers to seeking mental health services within their own congregations and to determine specifically what those barriers may be. Additionally, it could be helpful for other churches to intentionally develop informal forms of support within the church congregation that are not only supportive to church members but can provide support to church leaders as well. As communities of caring, churches offer unique opportunities to provide culturally as well as spiritually relevant support to all who gather to worship there regardless of their role. The origins of mental health challenges can differ greatly across populations, but the importance of mental health as a strong foundation for a longer and more fulfilling life is universal. Incorporating the strengths of faith and community into mental health care provides supports that are culturally inclusive and meaningful to those who need them.

A uniform approach is ineffective, and funders and policymakers must recognize this when identifying and developing effective strategies. Funding for community-based organizations should be contingent upon their responsiveness to this reality. It is further hoped that the findings of this study will provide a greater understanding of the role that community based mental health agencies can provide to the KA community by learning about its specific needs. Understanding the cultural norms and the role of churches and church leaders provides community-based agencies with opportunities to partner with them in possibly new and previously untapped ways. Incorporating the strengths of faith and community into mental health care provides supports that are culturally inclusive and meaningful to those who need them.

## Figures and Tables

**Table 1 ijerph-21-01094-t001:** Socio-demographic characteristics (*N* = 243).

	Church Leaders	Church Members
	*N* (%) or *M* (*SD*)	*N* (%) or *M* (*SD*)
Age, years (19–90)	55.93 (16.18)	41.22 (SD = 19.96)
Gender		
Female	65 (59.1%)	82 (61.7%)
Male	45 (40.9%)	51 (38.3%)
Generation		
1.5/2nd	24 (21.8%)	76 (57.1%)
1st	86 (78.2%)	57 (42.9%)
Marital Status		
Not married	20 (18.2%)	80 (60.2%)
Married	90 (81.8%)	53 (39.8%)
Education		
Highschool or less	12 (10.9%)	12 (9.0%)
Some college or more	98 (89.1%)	121 (91.%)
Employment		
Not employed	57 (51.8%)	68 (51.1%)
Employed	52 (47.3%)	65 (48.9%)
Income		
Not comfortable	15 (13.6%)	32 (24.1%)
Comfortable	95 (86.4%)	101 (75.9%)
Health Insurance		
No	15 (13.6%)	14 (10.6%)
Yes	95 (86.4%)	118 (89.4%)
Health Status (2–5)	3.32 (0.94)	
Coping Strategies		
Dealt with it myself		
No	79 (71.8%)	76 (57.1%)
Yes	31 (28.2%)	57 (42.9%)
Prayed		
No	37 (33.6%)	55 (41.4%)
Yes	73 (66.4%)	78 (58.6%)
Ignored it		
No	104 (94.5%)	122 (91.7%)
Yes	6 (5.5%)	11 (8.3%)
Exercised		
No	89 (80.9%)	62 (46.6%)
Yes	21 (19.1%)	71 (53.4%)
Confided in pastor/spiritual leader		
No	100 (90.0%)	113 (85.0%)
Yes	10 (9.1%)	20 (15.0%)
Confided in family/friends		
No	62 (56.4%)	62 (46.6%)
Yes	48 (43.6%)	71 (53.4%)
Acculturation (0–37)	16.40 (10.28)	19.13 (10.80)
Religious Coping (12–25)	20.91 (2.89)	19.57 (*SD* = 3.88)
Resiliency (0–40)	28.63 (6.96)	27.56 (*SD* = 6.99)
Social Support (3–30)	18.14 (5.64)	18.43 *(SD* = 5.45)

**Table 2 ijerph-21-01094-t002:** Bivariate relationship with mental distress variables (*N* = 243).

	Church Leaders				Church Members			
	Anxiety		Depression		Anxiety		Depression	
	*M* (*SD*)	*t* or *r*	*M* (*SD*)	*t* or *r*	*M* (*SD*)	*t* or *r*	*M* (*SD*)	*t* or *r*
Age, years (24–84)		0.11		0.07		0.25 **		0.04
Gender								
Female	1.57 (0.55)	0.23	1.61 (0.46)	0.36	1.60 (0.52)	1.83	1.75 (0.44)	0.29
Male	1.54 (0.54)		1.57 (0.51)		1.43 (0.47)		1.73 (0.52)	
Generation								
1.5/2nd	1.47 (0.58)	−0.88	1.59 (0.52)	−0.04	1.45 (0.46)	−2.23 *	1.73 (0.47)	−0.41
1st	1.58 (0.55)		1.60 (0.47)		1.64 (0.55)		1.77 (0.47)	
Marital Status								
Not married	1.51 (0.58)	−0.42	1.68 (0.55)	0.83	1.46 (0.47)	−2.12 *	1.75 (0.47)	−0.03
Married	1.57 (0.55)		1.58 (0.46)		1.65 (0.55)		1.75 (0.48)	
Education								
Highschool or less	1.38 (0.42)	−1.23	1.31 (0.32)	−2.24 *	1.81 (0.62)	1.99 *	1.82 (0.69)	0.39
Some college or more	1.58 (0.56)		1.63 (0.49)		1.51 (0.49)		1.74 (0.45)	
Employment								
Not employed	1.59 (0.61)	0.64	1.64 (0.50)	1.15	1.59 (0.55)	1.26	1.78 (0.51)	0.75
Employed	1.52 (0.49)		1.54 (0.46)		1.48 (0.46)		1.71 (0.43)	
Income								
Not comfortable	1.73 (0.55)	1.32	1.71 (0.55)	1.02	1.61 (0.54)	0.92	1.91 (0.59)	2.31 *
Comfortable	1.53 (0.55)		1.58 (0.47)		1.51 (0.50)		1.70 (0.42)	
Health Insurance								
No	1.50 (0.45)	−0.44	1.58 (0.58)	−0.14	1.55 (0.46)	0.08	1.92 (0.49)	1.40
Yes	1.57 (0.57)		1.60 (0.47)		1.54 (0.52)		1.73 (0.47)	
Health Status (2–5)		−0.39 ***		−0.32 ***		−0.45 ***	(−)0.31 ***	
Coping Strategies								
Dealt with it myself								
No	1.48 (0.50)	−2.52 *	1.52 (0.45)	−2.61 *	1.53 (0.55)	−0.12	1.73 (0.52)	−0.34
Yes	1.77 (0.62)		1.78 (0.52)		1.54 (0.46)		1.76 (0.39)	
Prayed								
No	1.70 (0.63)	1.97	1.80 (0.53)	3.38 ***	1.53 (0.48)	−0.2	1.75 (0.44)	0.1
Yes	1.49 (0.50)		1.49 (0.42)		1.54 (0.53)		1.74 (0.49)	
Ignored it								
No	1.54 (0.54)	−1.25	1.57 (0.48)	−2.01 *	1.53 (0.52)	−0.98	1.74 (0.47)	−0.64
Yes	1.83 (0.68)		1.97 (0.44)		1.68 (0.37)		1.83 (0.45)	
Exercised								
No	1.59 (0.56)	1.21	1.60 (0.49)	0.49	1.49 (0.50)	−1.47	1.75 (0.49)	0.17
Yes	1.43 (0.50)		1.55 (0.46)		1.63 (0.52)		1.73 (0.43)	
Confided in pastor/spiritual leader								
No	1.57 (0.56)	0.65	1.62 (0.49)	2.59 *	1.55 (0.51)	0.48	1.77 (0.48)	1.26
Yes	1.45 (0.51)		1.33 (0.31)		1.49 (0.51)		1.63 (0.39)	
Confided in family/friends								
No	1.54 (0.59)	−0.32	1.62 (0.51)	0.74	1.59 (0.50)	1.17	1.75 (0.50)	0.07
Yes	1.58 (0.50)		1.56 (0.45)		1.49 (0.51)		1.74 (0.45)	
Acculturation (0–37)		−0.06		−0.03		−0.12		0.04
Religious Coping (12–25)		−0.15		−0.27 **		−0.09		−0.25 **
Resiliency (0–40)		−0.36 ***		−0.41 ***		−0.30 ***		−0.32 ***
Social Support (3–30)		−0.13		−0.19 *		−0.13		−0.15

Note: * *p* < 0.05; ** *p* < 0.01; *** *p* < 0.001.

**Table 3 ijerph-21-01094-t003:** Hierarchal linear regression analysis predicting distress scores for church leaders (*N* = 110).

	Anxiety		Depression	
	Step 1 ***	Step 2 ***	Step 1 ***	Step 2 ***
	β	β	β	β
Age	0.17	0.18	0.19	0.16
Generational Status	−0.04	−0.02	−0.12	−0.09
Marital Status	0.01	0.05	−0.11	−0.05
Educational Status	0.21	0.17	0.29	0.26 **
Income	−0.08	−0.1	−0.09	−0.60
Health	−0.42 ***	−0.34 ***	−0.37 ***	−0.24 **
Dealt with it Myself		0.25 **		0.21 *
Prayed		−0.60		−0.16
Ignored it		0.06		0.13
Talked to Pastor/Spiritual Leader		−0.01		−0.11
Religious Coping		0.09		−0.08
Resiliency		−0.22 *		−0.22 *
Social Support		0.01		−0.03
*R* ^2^	0.21	0.34	0.21	0.41
Adjusted *R*^2^	0.17	0.25	0.16	0.33
Δ*R*^2^		0.13		0.20
*F*-statistic	4.64	3.78	4.50	5.16

Note. * *p* < 0.05; ** *p* < 0.01; *** *p* < 0.001.

**Table 4 ijerph-21-01094-t004:** Hierarchal linear regression analysis predicting distress scores for church members (*N* = 133).

	Anxiety		Depression	
	Step 1 ***	Step 2 ***	Step 1 *	Step 2 **
	*β*	*β*	*β*	*β*
Age	0.15	0.19	0.02	0.03
Generational Status	−0.02	0.00	−0.03	−0.02
Marital Status	0.08	0.10	0.02	0.01
Educational Status	0.02	0.05	0.05	0.08
Income	−0.02	−0.01	−0.15	−0.15
Health	−0.42 ***	−0.36 ***	−0.30 **	−0.20 *
Dealt with it Myself		0.09		0.06
Prayed		0.07		0.15
Ignored it		0.11		0.03
Talked to Pastor/Spiritual Leader		−0.03		−0.12
Religious Coping		0.05		−0.23 *
Resiliency		−0.18		−0.21 *
Social Support		0.06		−0.01
*R* ^2^	0.24	0.20	0.12	0.23
Adjusted *R*^2^	0.20	0.20	0.08	0.15
Δ*R*^2^		0.04		0.11
*F*-statistic	6.59	3.60	2.83	2.72

Note. * *p* < 0.05; ** *p* < 0.00; *** *p* < 0.001.

## Data Availability

The raw data supporting the conclusion of the article will be made available by the authors on request.

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
