# Peer review of "Community-Based Mental Health Challenges and Implications: Examining Factors Influencing Distress and Help-Seeking Behaviors among Korean American Church Leaders and Members in Greater Los Angeles"

_ijerph, 2024, doi:10.3390/ijerph21081094_

Round 1

Reviewer 1 Report

Comments and Suggestions for Authors

Comments on the Quality of English Language

The quality of English is very good.  There were only minor lapses in the paper that would need correction.

Reviewer 2 Report

Comments and Suggestions for Authors

This study investigates the levels of mental distress and help-seeking behaviors among Korean-American church leaders and members in the Greater Los Angeles area. Using a cross-sectional quantitative approach, the authors have examined various factors such as socio-demographic variables, acculturation, religious coping, resilience, and social support. Data were collected through surveys conducted in seven Korean American churches of different denominations from November 2017 to February 2018. The study analyzes anxiety and depression using the Hopkins Symptoms Checklist and explores the use of formal and informal resources, physical health, religious coping, resilience, and social support. There is merit to this study.  It brings back what has often been ignored issues pertinent to the trajectory of migration acculturation and mental distress.  Some points for the authors to consider are detailed below.

TITLE

I think the current title is adequate (The Differential Impact of Mental Health Factors on Korean-American Church Leaders and Church Members in Southern California).  However, the authors could think about whether this aligns of the themes of the study ("Mental Distress and Help-Seeking Behaviors among Korean American Church Leaders and Members: A Cross-Sectional Study in the Greater Los Angeles Area")

ABSTRACT

 Clarify the sample size and demographic details (e.g., age, gender) of the participants surveyed.

 Mention the specific dependent variables measured in the survey.

 Provide the actual findings of the hierarchical logistic regression (e.g., odds ratios or significant predictors).

 Ensure consistent terminology throughout (e.g., use "mental distress" or "distress scores" consistently).

 INTRODUCTION

The objectives should be more explicit and followed throughout the manuscript.  At one point, we were told “Therefore, the purpose of this study was to explore the scope of anxiety and depression among Korean American church leaders and to assess whether the factors impacting the of church leaders differ from that of their church members. Later, we were told that this study evaluated the levels of ‘mental distress and help-seeking behavior' among Korean American church leaders and church members.  Please reconcile these conflicting views.

 I think that this stated aims (…to explore the extent of anxiety and depression among Korean-American church leaders and to assess whether the factors impacting the church leaders differ from those of their church members).  I think it should be expanded to accommodate all aspects of the study, including the use of formal and informal resources, physical health, religious coping, resilience, and social support.

METHOD

Provide more context on how the survey was tailored to the Korean American church population. Include information on any pilot testing or validation processes to ensure the cultural relevance and reliability of the instruments.  The surveys are available in both English and Korean, which accommodates the language preferences of the participants.

Could the authors check if they used a 10-item or 25-item Hopkins Symptom Checklist (HSCL)?  To my knowledge, HSCL-25 (Part I has 10 items for anxiety symptoms and Part II has 15 items for depression symptoms. Furthermore, in the 25-item, the cutoff score of 1.75 or higher is used to identify potential cases.  Is this the same 10-item version of HSCL?

Recruiting 243 participants from seven churches may not provide a sufficiently large sample for robust statistical analysis, particularly when breaking down into subgroups (e.g., church leaders vs. members). Otherwise, justify your sample size.

 Specify the types of bivariate analysis performed and why certain statistical tests (e.g., chi-square and t-tests) were chosen. Clarify the rationale behind the use of multivariate logistic regression and how the models were constructed, including the selection of covariates.

While mean imputation and constant imputation are mentioned, more sophisticated methods (e.g., multiple imputation) might handle missing data more effectively, preserving the variance and relationships in the data.

Regarding multicollinearity, the criteria for the retention of variables (correlated at 0.80 or higher) could be further detailed to explain the decision-making process for the retention and omission of variables.

For the outcome measurements, the authors used (i) Hopkins Symptoms Checklist (HSCL-10), (ii) Short Form Health Survey 12 (SF-12), (iii) Brief Religious Coping Scale, (iv) Conner-Davidson Resilience Scale (CD-RISC) and (v) Lubben Social Network Scale (LSNS-6).  These scales are widely used in the literature, and psychometric properties are known.  The authors have attempted to create internal validity in the present context.  This is very nice.  One minor inquiry: Short-form Health Survey 12 (SF-12 is often reported to screen quality of life (QoL).  I would assume that the authors may have focused on the subscale for perceived physical health that is linked to QoL.  This needs to be explained if my hunch is correct here.

Maybe create a separate heading for this (“The study was approved by the Loma Linda University IRB.”).  Any number assigned to the ethical approval? Furthermore, the authors need to include some of these (..” Participation in the study was voluntary and all participants provided their informed consent before participating in the investigation. Participants were assured of their right to withdraw from the study at any time without consequences. The confidentiality and anonymity of the participants was maintained throughout the study by assigning unique identifiers and securely storing data..”)

RESULT

The summary indicates that the sample population was "mostly female," but specific numbers or percentages are not provided. Including this information would give a clearer picture of the gender distribution.

Clarify the meaning of "1.5/2nd generation" for readers who may not be familiar with these terms. Provide a brief explanation.

Mention the percentage of the overall sample that was employed versus that that was not employed to give a clearer picture of the employment distribution.

For both church leaders and members, provide the exact percentages or numbers for age categories (e.g., 19-30, 31-40, etc.) to better illustrate the age distribution.

Although percentages for anxiety and depression are provided above the cutoff scores, it would be helpful to include the actual number of participants these percentages represent (e.g., "30.5% of the sample population, or 74 participants, scored above the cut-off for anxiety").

The comparison between church leaders and members is noted, but the statistical significance of these differences should be reported if applicable (e.g., p-values or confidence intervals).

Explain the rationale for using the international cut-off score of 1.75 for the Hopkins Symptoms Checklist and provide a brief description of what this score indicates in terms of symptom severity.

DISCUSSION

While the study notes that religious coping was significantly associated with depression for church members but not leaders, it lacks a deeper exploration of why this difference exists. A more detailed analysis could provide insight into the different impacts of religious coping mechanisms in different roles within the church. There is a huge literature on this from which authors could draw inspiration.

The authors have alluded to the view of the potential of developing interventions based on roles within the church, but do not provide detailed strategies or examples. More specific recommendations for culturally aligned interventions would be interesting for readers.

Although the study mentions the need to identify barriers for church leaders seeking mental health services, it does not explore this in detail. A focused investigation of these barriers, including cultural stigma and availability of services, would strengthen the discussion section.

The study reports that social support was not significantly associated with distress for church members, but highlights its importance to church leaders. This discrepancy should be examined and explained more thoroughly to avoid confusion and provide a clearer understanding of the data.

LIMITATION

The authors acknowledge various limitations. Well done.  Perhaps the authors could consider 2 more: (I) The announcement of recruitment during church services can lead to self-selection bias, where only those interested in mental health issues participate. (2) A subtle issue that emerged from this study is that the study did not find significant differences in distress levels between age, gender, or generational status at the multivariate level. It is necessary to consider the complex interplay between these variables for a more complete understanding in future studies.

REFERENCE

The authors have employed 49 citations. Most of them are recent and relevant. There is only one indication of self-citations (J Immigr Minor Health . 2021 Jun;23(3):528-535. doi: 10.1007/s10903-020-01050-1.). This is well below the threshold.

Round 2

Reviewer 2 Report

Comments and Suggestions for Authors

The esteemed authors have revised or satisfactory rebutted my comments and suggestions. On this ground I have no hesitation to recommend this manuscript for publication. 
